# Towards precision psychiatry: Metabolomics identifies three biological subtypes of depression

**Simeng Ma**[1‡], **Zhaowen Nie**[1‡], **Mengyuan Zhang**[2‡], **Junhua Mei**[3], **Enqi Zhou**[1], **Zhiyi Hu**[2], **Honggang Lv**[1], **Qian Gong**[1], **Gaohua Wang**[1], **Huiling Wang**[1], **Bo Du**[4], **Jun Yang**[2◉*], **Zhongchun Liu**[1,5,6◉*]

1 Department of Psychiatry, Renmin Hospital of Wuhan University, Wuhan, China, 2 School of Information Engineering, Wuhan University of Technology, Wuhan, China, 3 Department of Neurology, Wuhan First Hospital, Wuhan, China, 4 School of Computer Science, Wuhan University, Wuhan, China, 5 Taikang Center for Life and Medical Sciences, Wuhan University, Wuhan, China, 6 State Key Laboratory of Metabolism and Regulation in Complex Organisms, College of Life Sciences, Wuhan University, Wuhan, China

‡ These authors are co-first authors on this work.
◉ These authors contributed equally to this work.
* zcliu6@whu.edu.cn (ZL); junyang_ie@whut.edu.cn (JY)

## Abstract

Depression is clinically and biologically heterogeneous, mandating classification strategies for personalized medicine. This study explored depression subtypes using metabolomics data from the UK Biobank and validated the subtypes in the Whitehall II cohort. The five-step analysis included: (1) identification of distinct subtypes using non-negative matrix factorization (NMF) and four machine learning algorithms; (2) genome-wide association studies (GWAS) to examine associations across subtypes and controls; (3) comparison of clinical characteristics across subtypes; (4) development of 24 subtype-specific diagnostic models and validation in an independent cohort; and (5) construction and comparison of metabolic networks across subtypes. Cluster analysis of 249 metabolomic indicators in individuals with current depressive episodes (n = 7,945) identified three metabolic subtypes of depression. Subtype 1 was characterized by fatty acid dysregulation, subtype 3 had a hyperlipidemia phenotype, while subtype 2 displayed an intermediate phenotype. Metabolic subtypes were not associated with SNPs. Diagnostic models built using the 249 metabolic indicators yielded the area under the curve (AUC) of 0.644 for the total depression sample and 0.785, 0.817, and 0.942 for subtypes 1, 2, and 3, respectively. Twenty-three additional diagnostic models based on combinations of metabolic indicators improved performance by 12.8-39.6% over a binary classification model. Metabolic networks significantly differed between each subtype and healthy controls but not between the total depressed group and controls. This study defines distinct metabolic subtypes of depression. Future research should combine high-throughput metabolomics with prospectively established depression cohorts and tailored interventions to explore subtype-specific diagnostic and therapeutic biomarkers.

**Data availability statement:** This research has been conducted using the UK Biobank Resource under Application Number 87530. Researchers can apply for data access via the UK Biobank Access Management System: https://biobank.ndph.ox.ac.uk/ukb/label.cgi?id=220. The Whitehall II Data was provided by Dementia Platforms UK (DPUK). All analyses were performed using the DPUK analysis portal. Researchers can explore and apply for access to this dataset through the DPUK Data Catalogue: https://data.dpuk.ukserp.ac.uk/CohortDirectory/Item?fingerPrintID=Whitehall%20II. The analytic code used in this study is fully available at: https://github.com/xiaogahuo/NMR_MDD_Subtype_Clustering.git.

**Funding:** This work was supported by grants from the National Natural Science Foundation of China (Grant No. 823B2031 to SM) and the National Key Research and Development Project of China (Grant No. 2024YFC3308400 to ZL). The funders had no role in study design, data collection and analysis, decision to publish, or preparation of the manuscript.

**Competing interests:** The authors have declared that no competing interests exist.

## Author summary

Depression is clinically and biologically heterogeneous, mandating classification strategies for personalized medicine. Can metabolomic profiling identify biologically distinct subtypes of depression and improve diagnostic accuracy? This study explored depression biological subtypes using metabolomics data. Analysis of 249 metabolic markers in 7,945 depressed patients using non-negative matrix factorization and machine learning revealed three subtypes: Subtype 1 (fatty acid dysregulation), Subtype 3 (hyperlipidemia), and Subtype 2 (intermediate phenotype). Diagnostic models incorporating these metabolic markers, validated in the Whitehall II cohort, outperformed binary classification, with AUC improvements of 12.8-39.6%. This study identifies three biological subtypes of depression, each demonstrating unique dysregulation patterns. Machine learning models incorporating these metabolic s indicators show enhanced diagnostic accuracy. These findings provide clues for future development of precision diagnostics and therapeutics for depression.

## 1. Introduction

Depression, a complex disease influenced genetic, environmental, psychological, and biological factors [1], is primarily diagnosed based on subjective clinical symptoms [2,3]. Despite progress in precision medicine, the absence of objective biomarkers for depression hampers clinical outcomes. Potential biomarkers for depression could originate from various dysregulated biological processes, including alterations in brain structure and function and peripheral changes in inflammatory, neurotransmitter, neurotrophic, neuroendocrine, and metabolic systems [1]. However, the complex etiology of depression cannot be fully captured by any single biological factor, and the interplay between these processes suggests that isolated examination may not suffice for clinical advancement.

Omics-based research, particularly metabolomics, offers a promising approach by capturing the intricate interactions between an organism and its environment [4]. Given the heterogeneity of depression, a binary diagnostic approach cannot accurately classify all patients. Instead, subtyping patients based on molecular or phenotypic data could enhance diagnosis, prediction, and treatment [5]. The metabolome, being the closest to the patient phenotype, is particularly suitable for molecular phenotyping [6].

Despite multiple metabolic alterations having been identified in depression [7–11], their utility as diagnostic biomarkers remain limited. Our previous machine learning analysis of the UK Biobank (n = 123,459) identified metabolic biomarkers associated with depression [12]. Diagnostic models incorporating these metabolites along with traditional risk factors achieved AUCs of 0.658 and 0.716 for lifetime and current depression, respectively. Here we build on this study to define depression subtypes based on the UK Biobank, the largest metabolomics database at present,

and validate them in the Whitehall II cohort. We perform a five-step analysis: (1) identification of distinct subtypes using non-negative matrix factorization (NMF) and four machine learning algorithms; (2) genome-wide association studies (GWAS) to examine associations across subtypes and controls; (3) comparison of clinical characteristics across subtypes; (4) development of 24 subtype-specific diagnostic models and validation in an independent cohort; and (5) construction and comparison of metabolic networks across subtypes.

## 2. Results

### 2.1. Demographic characteristics

The detailed data analysis process is illustrated in Fig 1. Details of the metabolomic indicators are shown in Table A in S2 File. Of 183,926 participants included in the study, 7,945 were diagnosed with current depression. Table B in S2 File presents the demographic and clinical characteristics of the two groups.

 A correlation heatmap of metabolomics data for individuals with current depression and healthy controls (HC) is shown in Fig 2A, which reveals strong correlations between lipoprotein subclasses and between various lipoprotein components. Compared to HC, individuals with current depression exhibited significantly more correlations between metabolic indicators (Fig 2B).

### 2.2. Metabolic subtypes of depression

By analyzing 249 metabolic features, we identified three distinct metabolic subtypes (Table C in S2 File). As shown in Table B in S2 File, the three subtypes of depression included 3086, 3379, and 1480 individuals, respectively. Compared to subtype 1, subtypes 2 and 3 contained higher proportions of obesity, males, a greater number of co-morbid chronic diseases and metabolism-related disorders. However, no significant differences were observed among the three subgroups in terms of immune-related diseases.

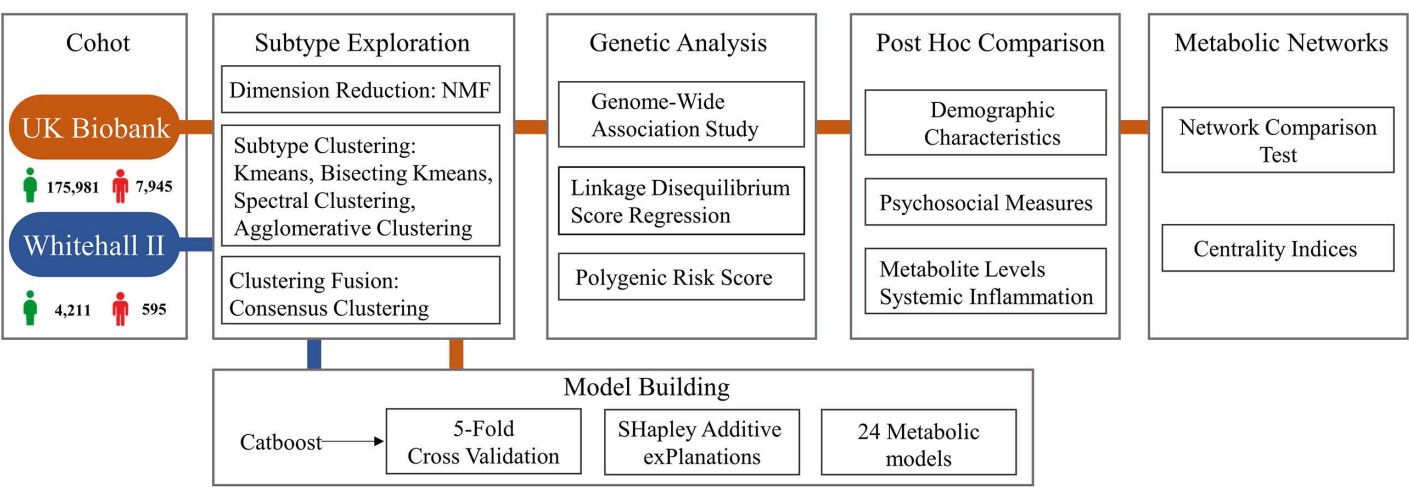

**Fig 1. Data analysis process.** Participants with depression were defined based on fields 130895 and 130896, which indicated the source and initial diagnosis date of depression, respectively. Exclusion criteria for the depression cohort included absence of metabolomics data, history of schizophrenia or bipolar disorder, diagnosed with cancer/ cerebrovascular disease/ substance dependence, missing depression diagnosis date, diagnosed with depression after the baseline assessment, and a PHQ-2 score <2 (indicating no current depressive episode). Healthy controls (HC) were included if they had metabolomics data, no psychiatric diagnosis, and a PHQ-2 score < 2.

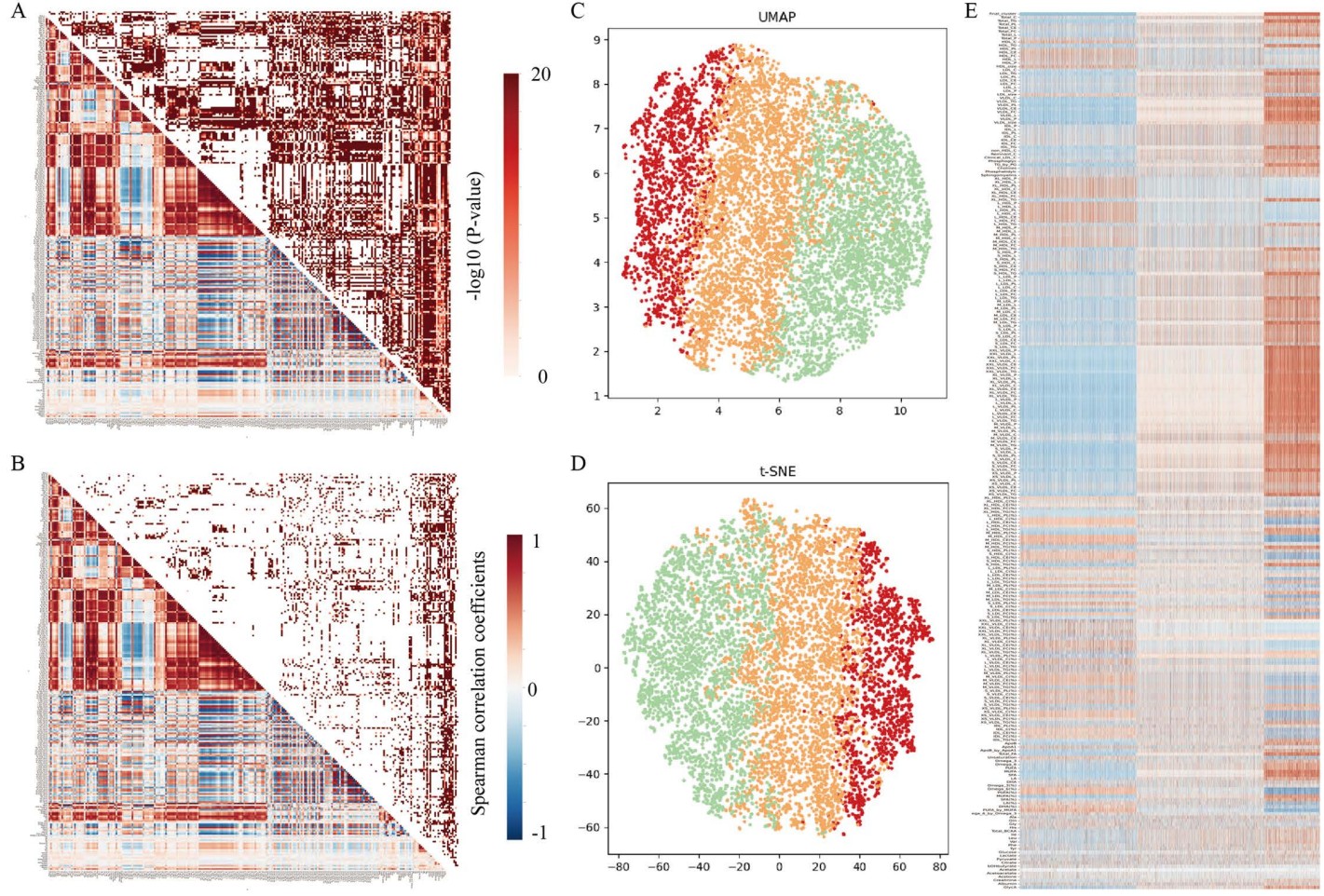

**Fig 2. Application of metabolomics data to identify depression subtypes.** Correlation heatmap of metabolomic data for individuals with current depression **(A)** and healthy controls **(B)**. **(C)** UMAP dimensionality reduction and visualization of identified clusters. **(D)** t-SNE dimensionality reduction and visualization of identified clusters. **(E)** Heatmap showing differential metabolites between the identified depression subtypes. In both A and B, correlations are shown in the bottom left, while the corresponding P-values are shown in the top right. These heatmaps represent -log10 transformed P-values for correlations between metabolic indicators.

The metabolomic differences between the three subtypes are depicted in Fig 2C and Table D in S2 File. The differences were particularly marked between subtypes 3 and 1, especially with respect to lipoprotein subclasses and fatty acid metabolism.

Subtype 3 showed upregulation of components associated with low density lipoprotein (LDL), very low density lipoprotein (VLDL), and intermediate density lipoprotein (IDL) subclasses, while high density lipoprotein (HDL) related components (excluding triglycerides) were downregulated. Subtype 1 showed completely opposite trends in these components. Similar trends were observed for fatty acid components: total fatty acids, omega-3, omega-6, polyunsaturated fatty acids (PUFA), monounsaturated fatty acids (MUFA), saturated fatty acids (SFA), and linoleic acid (LA) were significantly downregulated in subtype 1 and significantly upregulated in subtype 3. The degree of unsaturated PUFA to MUFA ratio (PUFA by MUFA) and docosahexaenoic acid to total fatty acids percentage (DHA%) were also significantly upregulated in subtype 1 and significantly downregulated in subtype 3.

Notably, subtype 3 exhibited characteristics consistent with hyperlipidemia, including elevated total cholesterol, tri-glycerides, and LDL cholesterol, as well as reduced HDL cholesterol, accompanied by increased apolipoprotein B, decreased apolipoprotein A1, and elevated apolipoprotein B to apolipoprotein A1 ratio.

Based on these results, we categorized the subtypes as follows: depression with fatty acid dysregulation (subtype 1), depression with an intermediate phenotype (subtype 2), and depression with hyperlipidemia (subtype 3).

## 2.3. GWAS results

GWAS data were present for 105,044 samples and 5,410,382 SNPs after quality control. We conducted four GWAS analyses to compare patients with depression (all depression cases and the three metabolic subtypes) with HC to explore the genetic basis of these subtypes. The results are shown in Fig 3A, and *QQ*-plots are provided in Fig A in S1 File.

No genome-wide significant SNP associations were identified. Further linkage disequilibrium score regression (LDSR) analyses were conducted to assess genetic correlations between depression subtypes and 34 diseases/traits, as summarized in Table E in S2 File. There were significant genetic correlations between the total depression sample and the three metabolic subtypes and broad depression, neuroticism, and subjective well-being ($P_{FDR} < 0.05$). The total depression sample and subtypes 1 and 2 showed significant genetic correlations with anxiety disorder, schizophrenia, bipolar disorder, anorexia nervosa, and insomnia, while subtype 3 did not. Other traits significantly correlated with the total depression

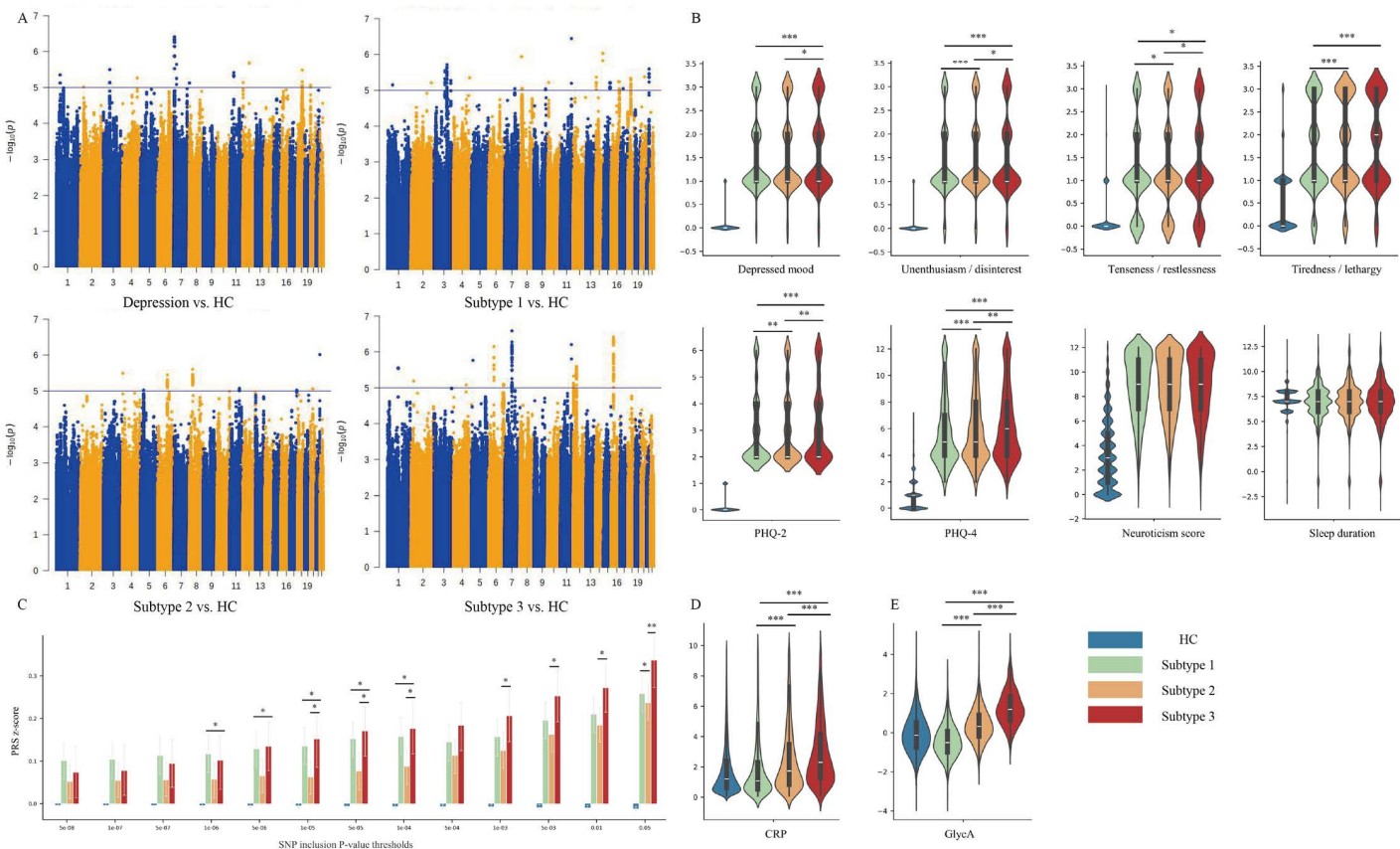

**Fig 3. Characteristics of identified metabolic subtypes. (A)** GWAS comparing HC with the total depression sample and three subtypes. **(B)** Comparison of depressive symptom among subgroups. **(C)** Comparison of polygenic risk scores among subgroups. **(D)** Comparison of C-reactive protein (CRP) levels among subgroups. **(E)** Comparison of GlycA levels among subgroups. * *P*<0.05, ** *P*<0.01, *** *P*<0.001.

sample, such as somatic pain, C-reactive protein (CRP), body mass index (BMI), visceral fat, lifestyle factors, and chronic diseases, showed differential patterns across subtypes. Notably, these genetic correlations were no longer significant in subtype 3.

### 2.4. Subgroup comparisons

We further compared depressive symptoms between the three metabolic subtypes (Fig 3B), observing a stepwise pattern (subtype 1 < subtype 2 < subtype 3) in Patient Health Questionnaire (PHQ)-2 scores, PHQ-4 scores, and four depressive symptoms. The subtype 2 polygenic risk score (PRS) consistently demonstrated an intermediate phenotype between the other two subtypes across various PRS calculation thresholds (Fig 3C).

There were also significant associations between depression subtypes and inflammatory markers. As shown in Fig 3D and 3E, individuals with subtypes 2 and 3 exhibited significantly higher circulating CRP and glycoprotein acetyls (GlycA) compared with subtype 1 and HC. For a more comprehensive analysis, see Table F in S2 File.

### 2.5. Machine learning algorithm for diagnostic prediction

We next constructed a model using 249 metabolomic indicators and evaluated its diagnostic performance in distinguishing HC, the total depression cohort, and subtypes 1, 2, and 3 (Fig 4A). Using 5-fold cross-validation to validate model performance and as shown in Fig 4C, the AUC values of the model for distinguishing depression, subtype 1, subtype 2, and subtype 3 in the test set were 0.644, 0.785, 0.817, and 0.942, respectively. The top-20 important features output by the model for subtypes 1–3 are shown in Fig 4B, noting some overlap in important features between the different subtypes. The top 20 features for the three models included 34 metabolic indicators, of which 27 were related to lipoproteins, 5 to fatty acids, and GlycA. Notably, 18 features were related to triglyceride-rich lipoproteins (TRLs).

Given the significant differences in lipoprotein-related indicators between the different subtypes, we further constructed 23 additional models, as shown in Fig 4C. The metabolic indicators used in each model are listed in Table G in S2 File.

The diagnostic performance based on subtype classification demonstrated superior efficacy compared to the traditional binary classification approach, with an average increase in AUC ranging from 12.8% to 39.6% in the test set (Fig 4C). Specifically, the AUC for subtype 1 showed an improvement of 13.2% to 22.6%, for subtype 2 by 12.8% to 24.6%, and for subtype 3 by 20.6% to 39.6%. To further evaluate the robustness of the machine learning-based diagnostic performance, gender-stratified analyses were conducted. The results revealed that the AUC improved by 8.9% to 39.0% in female participants (Fig 5A) and by 1.9% to 37.3% in male participants (Fig 5B). Additionally, considering the potential influence of chronic diseases and BMI on metabolomics, sensitivity analyses were performed. After excluding participants with chronic diseases, the AUC increased by 11.9% to 40.5% (Fig 5C). Similarly, after excluding participants with abnormal BMI values, the AUC improved by 11.6% to 40.9% (Fig 5D).

As shown in Fig 6, to ensure the robustness of the results, four additional diagnostic models were constructed in the UK Biobank dataset. After sequentially excluding individuals with immune-related diseases (Fig 6A), metabolism-related diseases (Fig 6B), those taking chronic medications (Fig 6C), and those with all of the above conditions (Fig 6D), the AUC increased by 10.1–39.6%, 11.5–40.4%, 12.6–40.3%, and 13.6–41.9%, respectively.

In these analyses, the diagnostic performance of residual cholesterol, IDLs, and size was slightly worse, while models including TRL and different lipoprotein components performed better.

The robustness of the identified metabolic subtypes was validated in the independent Whitehall II cohort. As depicted in Fig 7, the classification model based on subtypes significantly outperformed the traditional diagnostic model in the Whitehall II cohort, with the AUC improvement ranged from 10.4% to 40.0%. Subgroup analyses further corroborated these results, showing AUC improvements of 8.2% to 42.7% in females (Fig 8A) and 9.4% to 40.5% in males (Fig 8B). In additional sensitivity analyses, after excluding participants with chronic diseases, the AUC improved by 11.9% to 39.1% (Fig 8C), and after excluding those with abnormal BMI, the AUC improved by 8.6% to 38.6% (Fig 8D).

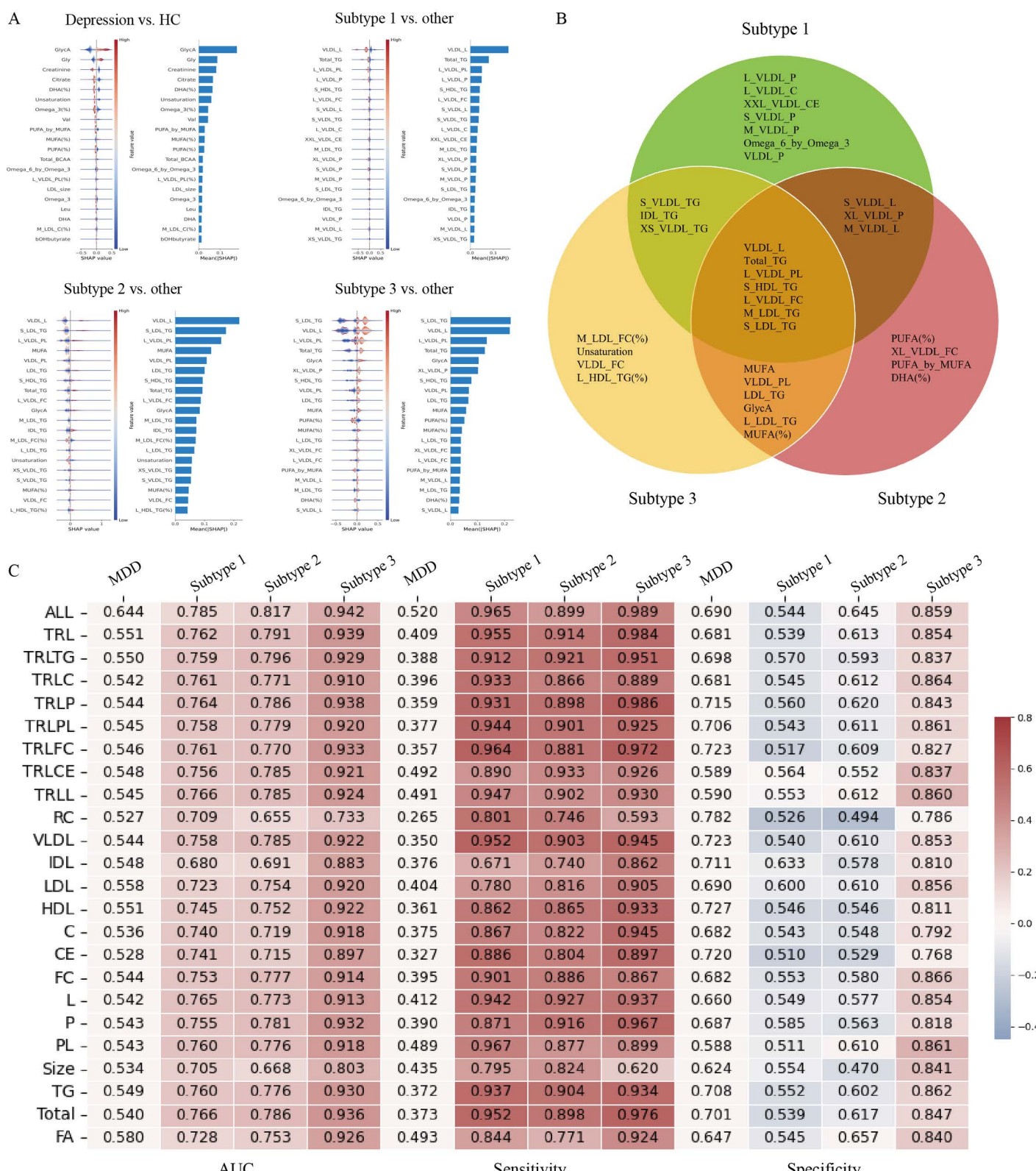

**Fig 4. Predictive performance of machine learning models for depression diagnosis. (A)** The top 20 important features identified by the models for the total depression sample and each subtype. **(B)** Venn diagram showing the overlap of important features among the three subtypes. **(C)** Performance of 24 machine learning models in the UK Biobank.

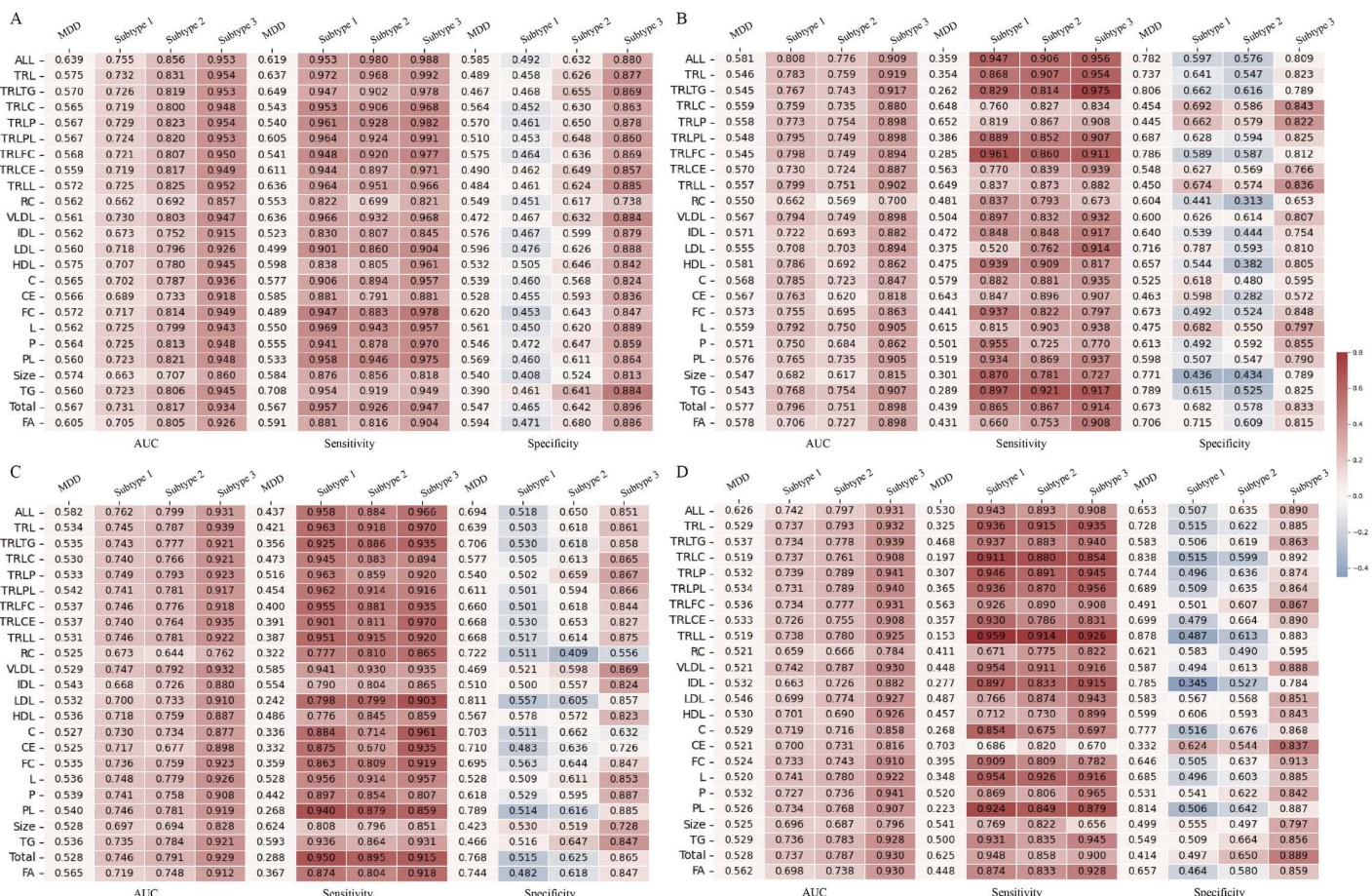

**Fig 5. Sensitivity analysis of the machine learning model's diagnostic performance in the UK Biobank dataset.** Stratified by: **(A)** Female participants. **(B)** Male participants. **(C)** Participants without chronic diseases. **(D)** Participants with normal BMI (18.5–30).

## 2.6. Comparison of metabolic networks between different subtypes

Given the complex interdependencies between metabolites, we applied network analysis to 34 machine learning-selected metabolic indicators in HC, total depression samples, and subtypes 1–3 to identify central metabolic indicators in different subgroups and to compare metabolic patterns across these subgroups. The resulting metabolic networks for different subgroups are visualized in Fig B1 in S1 File. Centrality indices within these networks are depicted in Fig B2 in S1 File. To further explore the influence of potential confounders, covariates such as sex, age, BMI, and the number of chronic diseases were included in each model (Fig 9A). Centrality indices within these networks are depicted in Fig 9B. Based on the expected influence metric, we screened the top three metabolic indicators for each subgroup (Table H in S2 File). A total of eight metabolites were identified: GlycA, triglycerides to total lipids in large HDL percentage [L_HDL_TG (%)], triglycerides in small HDL (S_HDL_TG), concentration of very large VLDL particles (XL_VLDL_P), cholesteryl esters in

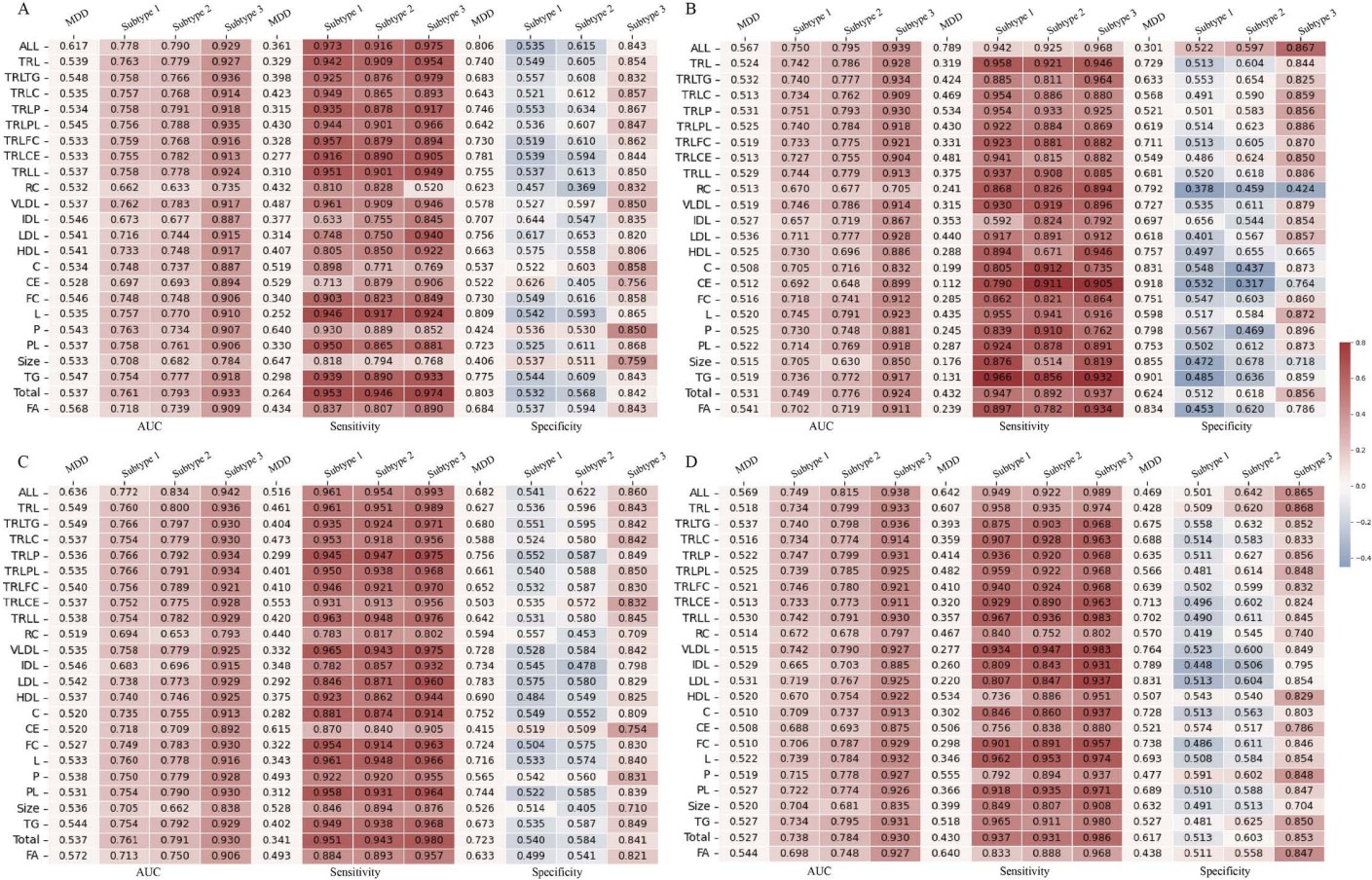

**Fig 6. Sensitivity analysis of the machine learning model's diagnostic performance in the UK Biobank dataset.** Stratified by: **(A)** Participants without immune-related diseases. **(B)** Participants without metabolism-related diseases. **(C)** Participants without taking chronic medications. **(D)** Participants without all of the above conditions.

chylomicrons and extremely large VLDL (XXL_VLDL_CE), MUFA, MUFA (%), and PUFA (%). Except for PUFA (%), the remaining seven metabolites showed consistent trends across the three depression subtypes, being lower in subtype 1 and higher in subtype 3. PUFA (%) exhibited an opposite trend.

Network stability and accuracy were deemed satisfactory. We examined differences in global connectivity and network structure between different depression subtypes and HC. As anticipated, there were significant network connectivity and structure differences between different subtypes and HC but not between the overall depression group and the HC group (P > 0.05). Detailed metrics are presented in Fig 9B.

## 3. Discussion

Here we applied cluster analysis to 249 metabolomic indicators in a large cohort of individuals with current depressive episodes. In doing so, we identify three metabolic subtypes of depression. Subtype 1 is characterized primarily by fatty acid dysregulation and subtype 3 by hyperlipidemia. Subtype-based diagnostic models performed significantly better than a binary classification model, with an average increase in AUC ranging from 12.8% to 39.6% in the test set. The results of gender-stratified analyses, sensitivity analyses, and validation in an independent cohort collectively underscore the robustness of the identified metabolic subtypes.

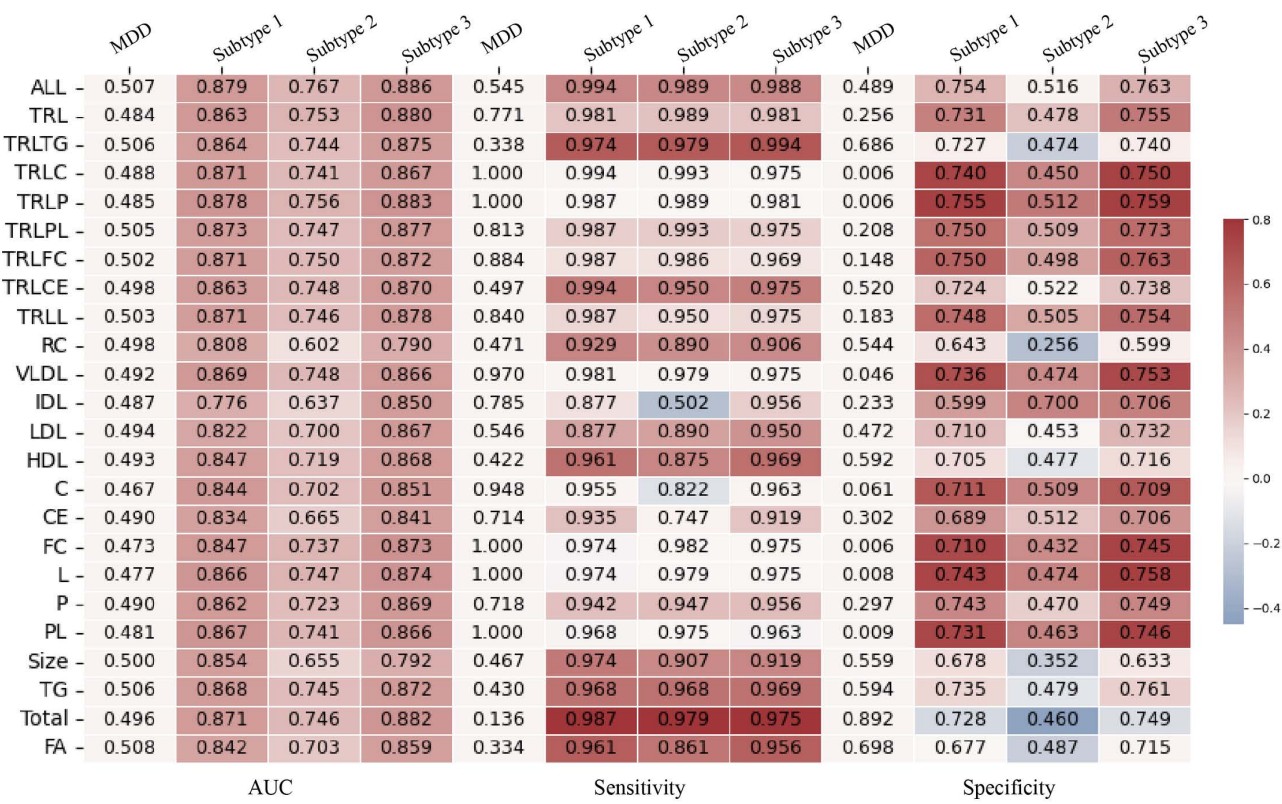

**Fig 7. Performance of machine Learning models in predicting depression in the Whitehall II cohort.**

Subtype-based analysis suggests the presence of abnormal metabolic networks in depression, which may be masked by its heterogeneity, especially when using a binary classification approach.

Over the past decade, many studies have investigated the relationship between depression and metabolic dysregulation [12,13]. Metabolic syndrome is present in up to one-third of all individuals with depression, although the reported prevalence varies across studies [14–17]. Moreover, 20–30% of individuals with depression exhibit immunometabolic dysregulation [18]. This immunometabolic depression is characterized by atypical, energy-related depressive symptoms, systemic low-grade inflammation, and metabolic abnormalities [19]. In our study, subtype 3 demonstrated a phenotype indicative of immunometabolic dysregulation, characterized by hyperlipidemia and elevated peripheral inflammatory markers. However, genetic correlation analysis did not reveal significant associations with CRP or chronic diseases. This finding supports a polygenic architecture model, suggesting that this dysregulation may be associated with genes of small effect size, without any dominant genetic variants [19] and more likely attributable to external environmental changes, including inflammation and chronic diseases. Targeted interventions for inflammation, metabolism, or lifestyle in this homogenous group of individuals with depression may be effective treatment options [18,20,21].

The metabolic profile of subtype 3 is consistent with clinical hyperlipidemia. Although dyslipidemia is associated with various factors, including chronic brain injury, aging, and mental health, the precise pathophysiological mechanisms remain incompletely understood [22–24]. One plausible mechanism involves cholesterol's modulation of neurotransmitter signaling, specifically affecting serotonergic, GABAergic, and glutamatergic neurotransmission, coupled with potential disruptions in synaptic plasticity and myelination, which may contribute to alterations in mental health [25]. As a key precursor for steroid hormone biosynthesis, cholesterol also plays a critical role in the

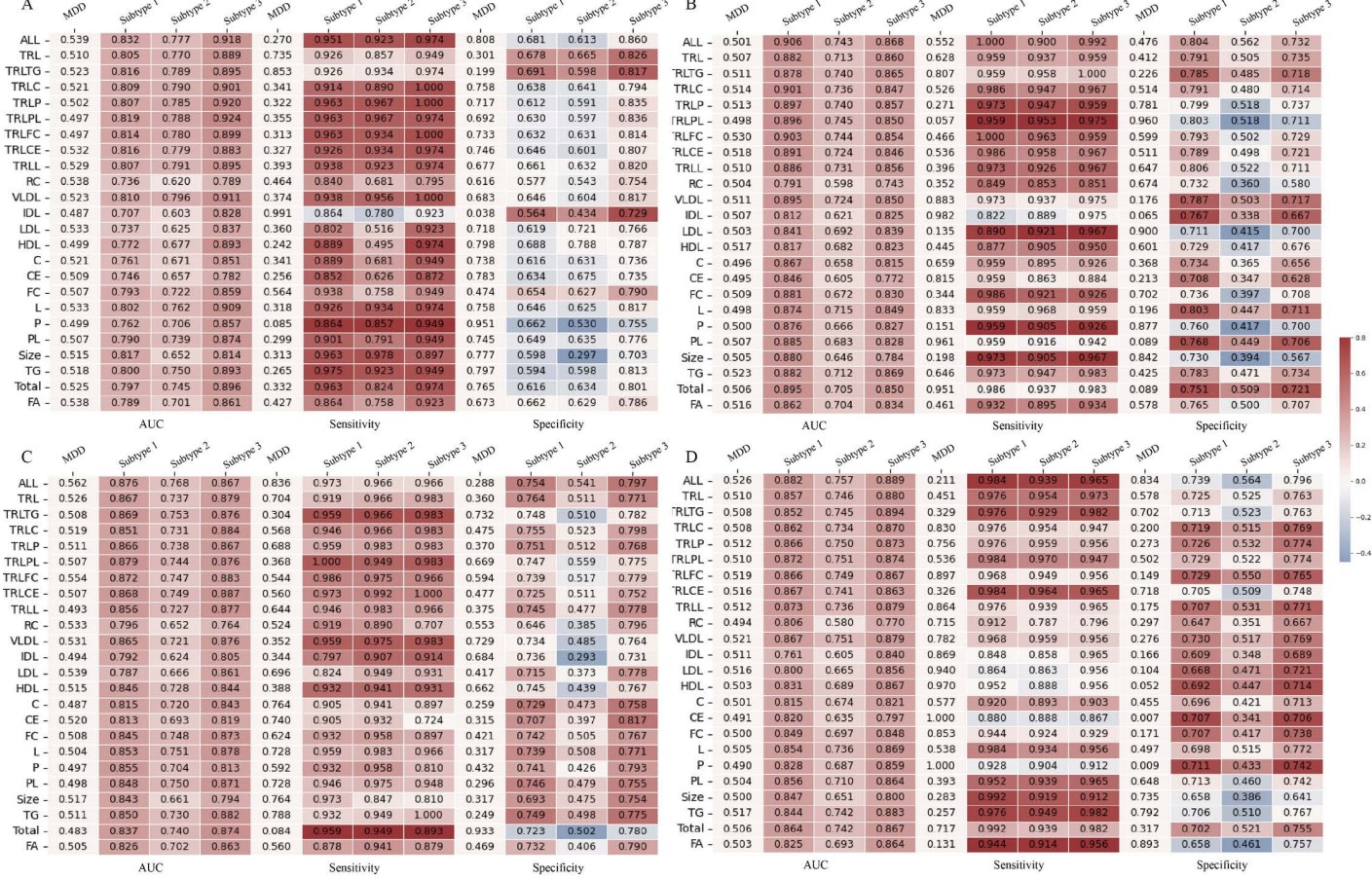

**Fig 8. Sensitivity analysis of the machine learning model's diagnostic performance in the Whitehall II cohort.** Stratified by: **(A)** Female participants. **(B)** Male participants. **(C)** Participants without chronic diseases. **(D)** Participants with normal BMI (18.5–30).

hypothalamic-pituitary-adrenal axis, a crucial regulator of circadian rhythms, stress responses, and neuropsychiatric disorders [25]. Furthermore, elevated circulating lipids may compromise blood-brain barrier integrity, potentially promoting neuroinflammatory processes that could impact mental health. Critically, however, the blood-brain barrier effectively restricts the transport of cholesterol-rich lipoproteins into the brain parenchyma, thereby limiting the direct influence of circulating cholesterol on neuronal function in the context of an intact blood-brain barrier [24]. This observation is a key consideration when postulating a causal link between systemic hypercholesterolemia and brain pathology. Therefore, further investigation of the precise causal relationship between hyperlipidemia and depression and the specific pathophysiological mechanisms involved is warranted. While there is increasing evidence of a potential role for cholesterol in the pathophysiology of depression, its feasibility as a therapeutic target remains an open question. Exploiting this approach is clearly not a one-size-fits-all solution for depression, as patients in this study with subtype 1 did not exhibit any abnormalities in cholesterol or other lipoproteins. A lipoprotein-targeted intervention appears more appropriate for subtype 3, highlighting the need for personalized treatment approaches. Clinical research incorporating metabolomics is necessary to clarify this point.

Previous research has established a link between dysregulation of fatty acid metabolism and depression, with PUFA receiving the most attention. These essential fatty acids, crucial for normal brain development, are found in dietary

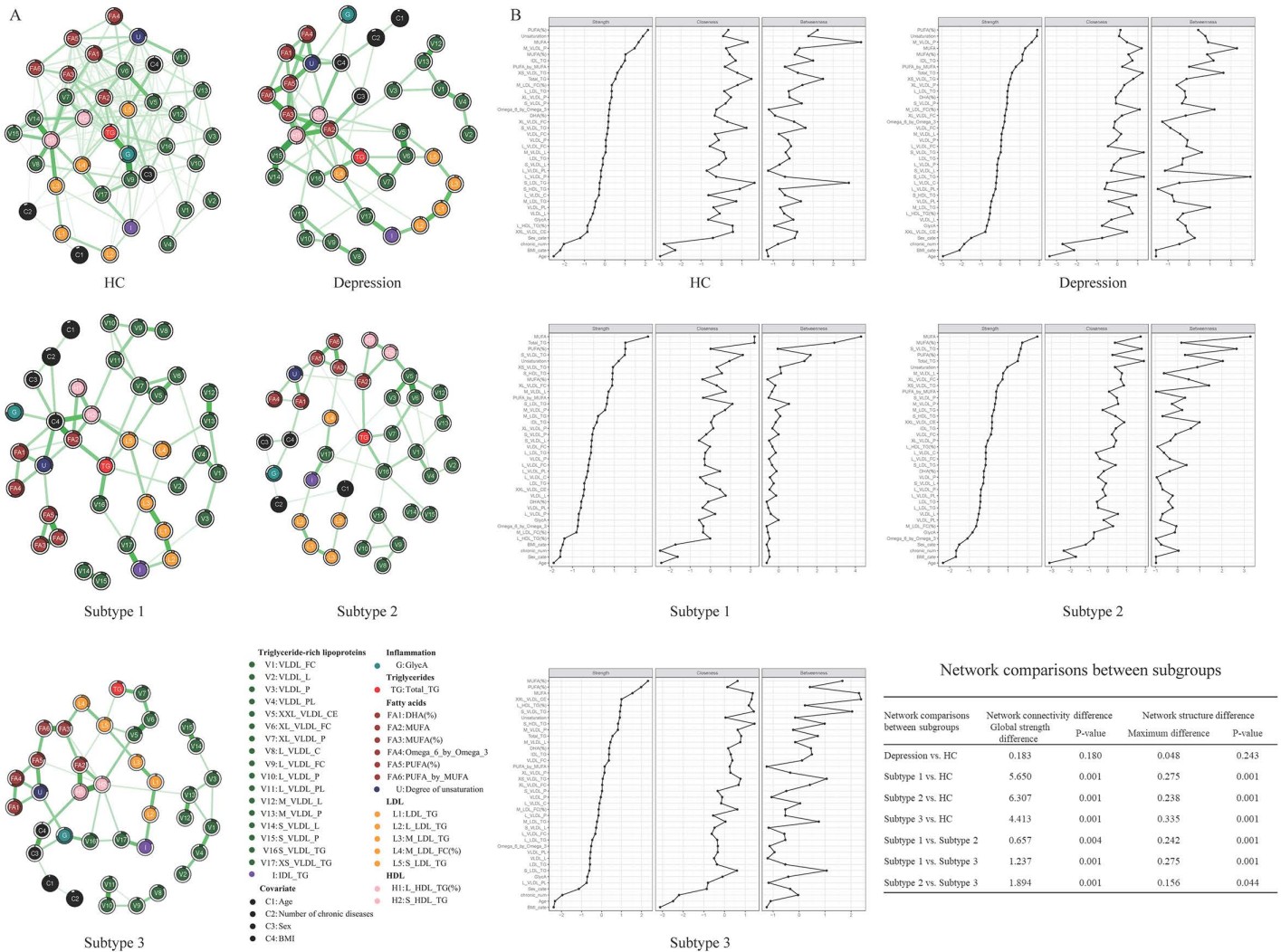

**Fig 9. Comparison of metabolic networks across depression subgroups. (A)** Metabolic networks for each subgroup. **(B)** Comparison of centrality indices, global connectivity, and network structure among subgroups.

sources such as fish (omega-3 PUFAs) and vegetable oils (omega-6 PUFAs) [26]. Depleted levels of these lipids have been strongly implicated in the pathophysiology of depression, potentially through mechanisms involving dysregulated inflammatory responses, reduced antioxidant capacity, and disruptions in neurotransmission [27]. It has been shown that omega-6 PUFAs are pro-inflammatory, whereas omega-3 PUFAs exert anti-inflammatory effects [28]. This balance of inflammation, mediated by eicosanoid derivatives, serves as a signaling mechanism in both the central and peripheral nervous systems, regulating inflammatory processes. Some studies have reported decreased levels of omega-3 PUFAs in patients diagnosed with depression, prompting numerous clinical trials to examine the effects of various omega-3 PUFA supplementation strategies on depression [29–31]. However, the findings from these investigations have been inconsistent, potentially due to relatively small effects being masked by heterogeneity. A more promising avenue lies in personalized medicine approaches targeting specific subgroups. In this study, subtype 1 exhibits a metabolic profile primarily characterized by fatty acid dysregulation, suggesting that omega-3 PUFA intervention may be particularly efficacious in

this subgroup. Our findings provide valuable insights for future clinical trial design, advocating pre-trial participant stratification based on metabolomic profiling to enable personalized therapeutic strategies tailored to distinct metabolic profiles.

Although the robustness of our model was maintained after adjusting for multiple potential confounders—including sex, BMI, chronic diseases, and medication use—it remains unclear whether the biological subtypes identified in this study, particularly the hyperlipidemia subtype, primarily reflect metabolic consequences of behavioral factors (such as sedentary behavior or poor dietary patterns) rather than depression-specific biological mechanisms. This important distinction warrants further investigation. Notably, the observed significant elevation of inflammatory markers within this subtype suggests that both biological and behavioral factors may interact in a bidirectional manner. Specifically, depressive symptoms may promote behavioral changes that exacerbate metabolic dysfunction, while metabolic alterations may in turn aggravate depressive symptoms through disruptions in energy homeostasis and neuroendocrine regulation. Given these complex interactions, future studies should explore whether tailored interventions—such as physical activity and dietary modifications—could serve as effective adjunctive therapies for specific depression subtypes, potentially mitigating both metabolic and mood-related symptoms.

Network analysis is a useful means to understand and visualize the heterogeneous nature of depression [32–34]. To further explore our identified biological subtypes, we employed network analysis to construct metabolic networks for each subgroup. Interestingly, the metabolic network of the overall depression sample was not significantly different to HC. However, the networks of our identified subtypes differed significantly from HC and from each other, further demonstrating the existence of distinct metabolic network patterns within depression. A recent analysis of data from the Netherlands Study of Depression and Anxiety (N = 2498) investigated associations between 30 depressive symptoms and 46 metabolites, finding that the somatic symptoms of fatigue and hypersomnia, along with cholesterol and fatty acids, were central nodes in the network [35]. Drawing on this research, establishing a comprehensive network of individual depressive symptoms and metabolomics could facilitate the discovery of metabolic networks strongly associated with specific symptoms, thereby enabling targeted interventions [32]. However, given the limitations of the current data, which included only four depressive symptoms, we did not incorporate symptoms in our analysis.

The heterogeneity of depression has long impeded the discovery of biomarkers and the development of precise therapeutic strategies. This study demonstrates that stratifying patients into biologically homogeneous subtypes based on metabolomic profiles significantly enhances diagnostic precision. Unlike traditional binary classifications—which have shown limited utility in biomarker discovery—our approach identifies metabolically distinct subgroups, allowing for more accurate phenotypic differentiation. TRL and related lipoprotein components may help characterize specific metabolic subtypes of depression. As the terminal phase of systems biology research, metabolomics comprehensively reflects the dynamic changes of metabolites under pathophysiological conditions, thereby providing a critical biological basis for disease subtyping. Based on biologically defined subtypes derived from metabolomics, researchers can achieve finer classification of mental disorders and further integrate high-throughput multi-omics technologies—such as genomics, proteomics, and transcriptomics—to systematically elucidate underlying pathogenic mechanisms. This strategy is analogous to the clinical subcategorization of depression into subtypes such as "with anhedonia" and "without anhedonia," aiming to uncover the molecular foundations of various subtypes and advance the implementation of individualized treatment and precision psychiatry. Future studies could explore whether these metabolic features contribute to subtype-specific diagnostic or intervention strategies, particularly when combined with other clinical and omics biomarkers. Indeed, metabolic disturbances are also commonly observed in other psychiatric disorders, such as schizophrenia, anxiety disorders, and bipolar disorder [36–38]. Data-driven approaches could similarly be applied to identify biologically distinct subtypes within these conditions and promote transdiagnostic research in psychiatry. Most importantly, these metabolically defined subgroups open new avenues for targeted and subtype-specific interventions. By aligning treatment strategies with specific dysregulated metabolic pathways, we move closer to the vision of precision psychiatry—where therapy is tailored to an

individual's biological subtype. This represents a critical step forward from traditional one-size-fits-all diagnostic and therapeutic models.

This study has several limitations that should be acknowledged. First, the cross-sectional design prevents us from establishing causal relationships between the identified metabolic subtypes and clinical outcomes. Although multiple sensitivity analyses were conducted, residual confounding from lifestyle factors—such as dietary habits and physical activity—cannot be fully ruled out. The observational nature of the data also limits our capacity to infer the direction of causality between metabolic dysregulation and depression. Future studies should adopt longitudinal designs and incorporate causal inference methods to verify the stability and biological foundations of these subtypes. Second, the scope of metabolomic coverage was limited by the number of metabolites available, which may affect the biological interpretability of the identified subtypes. We are currently addressing this issue in ongoing research through the use of high-throughput metabolomic platforms to uncover more precise and subtype-specific biomarkers. Third, although the UK Biobank currently offers the largest metabolomic dataset available, the dynamic characteristics of metabolic profiles necessitate careful consideration of disease state and sample collection timing. While the PHQ-2 was used for case identification, this instrument does not provide a comprehensive clinical assessment. Future work should employ more detailed clinical evaluations to better characterize subtype-specific symptomatology and its associations with metabolite profiles.

## 4. Methods

### 4.1. Data source and research design

The UK Biobank, a prospective cohort study of ~500,000 UK adults aged 40–69, provides a rich resource of genetic and phenotypic data [39]. The UK Biobank received ethical approval from the North West Multicenter Research Ethics Committee, with reference number 11/NW/0382. To explore the relationship between metabolites and depression, we selected individuals experiencing a current depressive episode from this cohort. Participants with depression were defined based on fields 130895 and 130896, which indicated the source and initial diagnosis date of depression, respectively. The exclusion criteria for the depression cohort comprised the following: absence of metabolomics data, a history of schizophrenia, bipolar disorder or cancer, missing date of depression diagnosis, diagnosis of depression after the baseline assessment, and a PHQ-2 score < 2 (indicating no current depressive episode) [40]. HC were included if they had metabolomics data, no psychiatric diagnosis and cancer, and a PHQ-2 score < 2.

### 4.2. Plasma biomarker profiling by NMR

Metabolomic profiling was conducted at Nightingale Health (Finland) on EDTA plasma samples from approximately 280,000 UK Biobank participants. A total of 251 metabolic markers were quantified using eight high-throughput spectrometers. To align with validation cohorts and previous studies, glucose-lactate and spectrometer-corrected alanine were excluded from the analysis. Details of the metabolomic indicators are shown in Table A in S2 File. The indicators cover a wide range of metabolic pathways, including 14 subclasses of lipoprotein lipids (210), fatty acids and fatty acid constituents (18), as well as various small molecule metabolites such as amino acids (10), ketone bodies (4), glycolytic metabolites (4), and those involved in fluid balance (2) and inflammation (1). Among these, 92 indicators are associated with TRLs, macromolecules composed of a large neutral lipid core (TG) and polar components including phospholipids, free cholesterol, and apolipoproteins. TRLs originate from the intestine and the liver and include chylomicrons, VLDLs and IDLs, and they serve as the primary source of fatty acids for energy production in peripheral tissues or storage in adipose tissue [41–43].

### 4.3. Determining the optimal number of subtypes

Following dimensionality reduction via NMF [44,45], the resulting features were visualized using both Uniform Manifold Approximation and Projection (UMAP) and t-Distributed Stochastic Neighbor Embedding (t-SNE) to illustrate underlying cluster

structures in a low-dimensional space. We performed clustering analysis on the metabolomics data of participants with current depressive episodes using four machine learning algorithms: *k*-means, bisecting *k*-means, spectral, and agglomerative clustering. The optimal number of subtypes was determined by systematically varying the number of dimensions and subtypes from 2 to 8 using three clustering evaluation metrics: silhouette coefficient, Calinski-Harabasz index, and Davies-Bouldin index.

### 4.4. Genome-wide association study analysis

Analysis in this study was conducted with version 3 of the UKBB imputed data, with 487,409 samples imputed and available for analysis following UKBB centrally performed QC filtering.

   After excluding participants of non-White ancestry, high missingness, relatedness, discordant genetic and self-reported sex, and filtering SNPs based on minor allele frequency (MAF) ≥ 0.05 and imputation INFO score > 0.9, 105,044 samples were retained for subsequent analyses. All association tests were performed in *PLINK* v1.90b6.21 using logistic regression. The analysis was adjusted for age, sex, genotyping batch, genotyping array, and 5 population principal components. Genetic correlations between depression subgroups and 34 diseases/traits were computed using LDSR analyses, implemented using the 'ldsc' package [46]. GWAS summary statistics used in this study are available at https://zenodo.org/records/10515792 and GWAS Catalog. P-values were adjusted for multiple testing using Benjamini-Hochberg false discovery rate (FDR) correction. This analysis is described in further detail in Supplemental Method A in S1 File.

### 4.5. Polygenic risk scores

To evaluate the cumulative genetic risk for depression, we calculated a PRS based on summary statistics derived from the iPSYCH depression GWAS, excluding data from the UK Biobank and 23andMe datasets [47]. As these summary statistics are based on people of European descent, we restricted PRS calculations to the European samples in the UK Biobank. Following Collister et al. [48], we calculated the PRS for depression across a range of SNP inclusion thresholds.

### 4.6. Network analyses

To explore the complex relationships between metabolites across different subgroups, we constructed metabolic networks using network analysis. This approach offers the advantage of identifying associations within the system while simultaneously considering the influence of all other variables [35,49].

   In network analysis, nodes represent variables, and edges represent the partial correlation coefficient between two nodes. Network analysis provides quantitative centrality indices for each node. Three common methods for measuring centrality indices are betweenness, closeness, and strength [50]: betweenness centrality measures the number of times a node lies on the shortest path between two other nodes; closeness centrality measures the average distance of a node to all other nodes in the network; and strength centrality represents the sum of the weights (e.g., correlation coefficients) of the edges connected to a node. We introduced expected influence (EI) to assess the nature and strength of the cumulative influence of nodes in the network. Centrality stability refers to the consistency of the ranking of centrality indices (e.g., betweenness, closeness, and strength) after re-estimating the network with fewer cases or nodes. Centrality stability was calculated using the correlation stability coefficient (CS-coefficient), which should ideally be > 0.5 to interpret centrality differences. 95% bootstrap confidence intervals (CIs) were used to estimate the accuracy of edge weights, where larger CIs indicate a lower accuracy of edge estimates and narrower CIs indicate a more reliable network. Finally, we examined differences in global connectivity and network structure between the metabolic networks of different subgroups.

### 4.7. Validation in an independent cohort

The Whitehall II cohort (WHII), a prospective study of 10,308 UK civil servants aged 35–55, has been followed up every 2–5 years since 1985 [51]. Given the availability of metabolomics data from phase 5, our analysis was restricted to

participants enrolled in phase 5 or earlier. Participants with current depression were selected using the same criteria as in our previous study: history of depression in phase 4 and a General Health Questionnaire depression subscale score ≥4 [12].

## 4.8. Statistical analysis

Common demographic characteristics and clinical variables were compared between different subtypes of depression and HC: sex, age, education level, BMI, economic status, physical activity level, smoking status, alcohol consumption, family history of depression, depressive symptoms, neuroticism, and number of chronic diseases (diabetes mellitus, heart diseases, and hypertensive diseases). Additionally, comparisons were made regarding the use of antidepressant medications, antipsychotic agents, medications for common chronic conditions, as well as the comorbidity of comorbid immune- and metabolism-related diseases across subgroups. A detailed description can be found in Supplementary Method B in S1 File.

Depressive symptoms were assessed using the first four items of the PHQ-9, focusing on frequency of depressed mood, unenthusiasm/disinterest, tenseness/restlessness, and tiredness/lethargy in the past two weeks. Neuroticism was assessed using an external 12-item scale derived from the Eysenck Personality Inventory neuroticism scale by Smith et al. [52], where higher scores (0–12) indicate greater neuroticism.

The CatBoost was used for diagnostic prediction, as previously [12]. We used 5-fold cross-validation and receiver operating characteristic (ROC) curves, sensitivity, and specificity to validate model performance. Shapley additive explanations (SHAP) were employed to assess the contribution of each feature to the model's predictive performance. To mitigate class imbalance, we incorporated class weighting within CatBoost by setting the class_weights parameter to values inversely proportional to the frequency of each class. For each subtype-specific model, a distinct control group was constructed using a one-vs-rest partitioning strategy, wherein cases of the target subtype were compared against all other participants (including other subtypes and non-depressed controls).

Given the significant differences in the prevalence of depression between genders, subgroup analyses were conducted separately for males and females. Additionally, considering the potential influence of chronic diseases and BMI on metabolomic profiles, sensitivity analyses were performed by excluding individuals with chronic diseases or abnormal BMI (<18.5 or ≥30). The same analytical procedures were subsequently replicated in the Whitehall II cohort.

Furthermore, to account for potential confounding effects of comorbid immune- and metabolism-related diseases, as well as the use of chronic medications (such as cholesterol-lowering medication, blood pressure medication, and insulin), we constructed four additional models to ensure the robustness of the findings. These models sequentially excluded participants with immune-related diseases, metabolism-related diseases, those taking chronic medications, and those with all of the above conditions.

Spearman correlation analysis was performed separately in the depression and HC groups. To compare variables between HC and different depression subtypes, we employed the Kruskal-Wallis test and chi-squared test for *post hoc* comparisons. This analysis investigated potential group differences across various domains, including demographic characteristics, psychosocial measures, PRS, and metabolite levels. In addition, we compared CRP and GlycA levels across subgroups to assess systemic inflammation. CRP is one of 30 blood biomarkers measured in the UK Biobank [53]. Statistical significance was set at a two-sided P-value threshold of 0.05.

All data were analyzed using Python v3.11. We utilized pandas v1.4 and numpy v1.26 for data preprocessing, while statistical analysis was conducted using scipy v1.12 and statsmodels v0.14. For machine learning, we employed scikit-learn v1.5, with experimental results depicted using matplotlib v3.8, seaborn v0.13. For network analyses, we used the R v3.6.3 package mgm v1.2.12 to estimate the network, qgraph v1.6.9 to visualize the network, bootnet v1.4.3 for stability and accuracy analysis, and Network Comparison Test package v2.2.1 for network comparison.

## Supporting information

**S1 File.** Supplemental Method A. Genome-wide association study analysis. Supplemental Method B. Supplementary methods for sensitivity analysis. Fig A. QQ plots of GWAS analysis for the total depression sample and three depression subtypes compared to healthy controls. Fig B. Comparison of metabolic networks across depression subgroups. (DOCX)

**S2 File.** Table A. 249 metabolomic indicators. Table B. Demographic Characteristics. Table C. Cluster evaluation index. Table D. The metabolomic differences between the three subtypes. Table E. Genetic correlations between depression subgroups and 34 diseases/traits. Table F. Comparison of the differences between different subgroups. Table G. The metabolic indicators used in each model. Table H. Centrality indices of different subgroups. (XLSX)

## Author contributions

**Conceptualization:** Simeng Ma, Gaohua Wang, Huiling Wang, Bo Du, Jun Yang, Zhongchun Liu.

**Data curation:** Simeng Ma, Jun Yang.

**Formal analysis:** Simeng Ma, Zhaowen Nie, Mengyuan Zhang, Zhiyi Hu, Jun Yang.

**Funding acquisition:** Simeng Ma, Zhongchun Liu.

**Methodology:** Simeng Ma, Enqi Zhou, Honggang Lv, Qian Gong, Gaohua Wang, Huiling Wang, Bo Du, Jun Yang.

**Project administration:** Jun Yang.

**Resources:** Jun Yang.

**Software:** Simeng Ma, Jun Yang.

**Validation:** Simeng Ma, Jun Yang.

**Visualization:** Simeng Ma, Jun Yang.

**Writing – original draft:** Simeng Ma.

**Writing – review & editing:** Zhaowen Nie, Mengyuan Zhang, Junhua Mei, Enqi Zhou, Zhiyi Hu, Honggang Lv, Qian Gong, Gaohua Wang, Huiling Wang, Bo Du, Jun Yang, Zhongchun Liu.

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
