## [Decision Letter · Decision Letter 0]

19 Aug 2025

Response to Reviewers
Revised Manuscript with Track Changes
Manuscript
**Journal Requirements:**
**Additional Editor Comments (if provided):**
**Reviewers' Comments:**

**Comments to the Author**

1. Does this manuscript meet PLOS Digital Health’s publication criteria?

Reviewer #1: Yes

Reviewer #2: No

Reviewer #3: Yes

2. Has the statistical analysis been performed appropriately and rigorously?

Reviewer #1: Yes

Reviewer #2: No

Reviewer #3: I don't know

3. Have the authors made all data underlying the findings in their manuscript fully available (please refer to the Data Availability Statement at the start of the manuscript PDF file)?

Reviewer #1: Yes

Reviewer #2: Yes

Reviewer #3: Yes

4. Is the manuscript presented in an intelligible fashion and written in standard English?

Reviewer #1: Yes

Reviewer #2: Yes

Reviewer #3: Yes

Reviewer #1: This is a well-designed and clinically relevant study that leverages metabolomic data from two large cohorts (UK Biobank and Whitehall II) to identify biologically distinct subtypes of depression. The identification of three metabolic subtypes—fatty acid dysregulated, hyperlipidemic, and intermediate—provides a valuable framework for advancing precision psychiatry. The study addresses a critical gap in depression research by moving beyond symptom-based classifications to biologically defined subtypes. This aligns with the NIH RDoC framework and could inform future targeted therapies. The study builds meaningfully on the authors’ prior work (Ma et al., 2024) by introducing clustering methods and improving diagnostic performance. While the work is strong, minor revisions would further strengthen its impact.

1. Please provide comparative analyses between depressed individuals and healthy controls (potentially as supplementary material) to assess whether these metabolic clusters are unique to depression or represent more general metabolic variations.

2. A discussion of whether similar clustering patterns occur in other psychiatric disorders (e.g., bipolar disorder, anxiety disorders) or metabolic conditions (e.g., metabolic syndrome, diabetes) would help contextualize the findings. Are these patterns more strongly associated with depression-specific pathophysiological mechanisms (e.g., neuroinflammation, HPA axis dysregulation)?

While the stability of results after adjusting for sex and BMI is commendable, please discuss whether the observed metabolic changes might reflect secondary consequences of depression-related behaviors (e.g., altered appetite, physical activity) rather than primary disease mechanisms. For example, could the "hyperlipidemic" subtype represent metabolic changes associated with sedentary behavior rather than depression-specific biology?

3. Please provide a more detailed characterization of the "intermediate" subtype. Is this a distinct biological group or more of a residual category? What are its defining metabolic or clinical features?

4. The manuscript should more explicitly highlight how the current clustering approach advances beyond the authors' previous machine learning analysis (Ma et al., 2024):

What specific improvements does the current methodology offer in terms of diagnostic accuracy, biological interpretability, or clinical utility?

Are there novel metabolic pathways identified in this study that were not apparent in the prior work?

5. The discussion should more carefully consider alternative interpretations of the metabolic findings: To what extent might the observed metabolic changes reflect lifestyle factors (e.g., diet, physical activity) rather than primary disease mechanisms?

While inflammatory markers (CRP, GlycA) and BMI differences are noted, please discuss whether these could be consequences rather than causes of the metabolic alterations.

These revisions would strengthen the manuscript's conclusions and provide a more nuanced interpretation of the important findings. The study represents a significant contribution to the field of precision psychiatry, and these suggested modifications would further enhance its impact.

Reviewer #2: In this manuscript, the authors attempt to identify subtypes of depression using plasma metabolomics data from the UK Biobank and subsequently build machine learning-based diagnostic models. While the goal of advancing precision psychiatry is commendable, this study suffers from several critical and fundamental flaws in its design, subject selection, and data analysis. These issues are pervasive and significantly undermine the validity of the findings. In my assessment, these flaws are not correctable through revision, and therefore, I must recommend the rejection of this manuscript.

Below are my detailed comments outlining these major concerns.

Major Concerns:

1. Critical Bias in Patient and Control Selection: The methodology for cohort selection is fundamentally flawed.

1.1 Heterogeneous Definition of Depression: The authors define cases using UK Biobank Data-Field #130895, which includes both formal ICD-10 diagnoses and self-reported depression. Self-reports are not equivalent to a clinical diagnosis of Major Depressive Disorder (MDD) and may simply reflect depressive symptoms secondary to other conditions.

1.2 Insufficient Exclusion Criteria: The manuscript states that the exclusion criteria include a history of schizophrenia or bipolar disorder. However, it fails to exclude patients with other major medical or psychiatric conditions that are known to profoundly alter metabolic profiles. For example, patients with a primary diagnosis of cancer, autoimmune disorders, or other chronic inflammatory diseases may be included in the depression cohort, making it impossible to disentangle the metabolic signature of depression from that of the comorbidity. The authors also did not exclude patients with anxiety disorders, a common comorbidity that shares metabolic and inflammatory pathways with depression.

1.3 Lack of Comorbidity Analysis: The study fails to stratify or at least compare the distribution of major comorbidities (e.g., cancer, cardiovascular disease) across the identified subtypes. This is a significant omission, as the "hyperlipidemia" subtype, for instance, could simply be a cluster of patients with metabolic syndrome rather than a true biological subtype of depression.

2. Lack of Temporal Alignment Between Diagnosis and Blood draw for Metabolomics: A crucial flaw is the failure to ensure that the metabolomics data were collected during a current depressive episode that was diagnosed prior to the blood draw. The Methods section notes an exclusion for those "diagnosed with depression after the baseline assessment," but the blood samples for metabolomics were collected over a wide window (2019-2022). The manuscript does not confirm that for each patient, the blood draw date was cross-referenced with the diagnosis date (Field #130896) to establish a clear temporal link. Without this confirmation, the metabolic state measured may precede the onset of depression or occur long after remission, rendering the entire analysis invalid for identifying state-dependent biomarkers.

3. Inherent Bias in Metabolomics Data: The study's conclusions are constrained by the severe limitations of the UK Biobank's metabolomics panel.

3.1 Limited and Biased Coverage: The panel measures only 249 metabolic indicators, which represents a tiny fraction (< 0.2%) of the known human metabolome (over 20,000).

3.2 Overrepresentation of Lipids: Critically, the panel is heavily biased towards lipid metabolism. As the authors note, 210 of the 249 markers are related to lipoprotein lipids and another 18 are fatty acids. With ~92% of the input data related to lipids, it is statistically inevitable that any clustering algorithm would identify subtypes based on lipid dysregulation. The study, therefore, does not discover novel biological subtypes so much as it confirms the input bias of its dataset. The "fatty acid dysregulation" and "hyperlipidemia" subtypes are likely artifacts of the measurement panel rather than distinct depression biotypes.

4. Uncontrolled Confounding Variables (For example the use of antidepressants and statins): The analysis fails to account for key confounders that directly influence metabolic profiles. Most notably, the use of antidepressant medications, statins, or other psychotropic and metabolic drugs was not compared between subtypes or controlled for in the analysis. These medications have well-documented effects on lipid, fatty acid, and inflammatory markers, and their differential use among patients could easily explain the metabolic differences observed between the proposed subtypes.

5. Flawed Machine Learning Methodology: The approach to building and evaluating the diagnostic models is unsatisfactory and lacks rigor.

5.1 Handling of Imbalanced Data: The dataset is extremely imbalanced, with 7,945 depression cases compared to over 175,000 healthy controls. The Methods section makes no mention of how this severe class imbalance was handled. Standard machine learning practice requires techniques like oversampling (e.g., SMOTE), undersampling (e.g., ENN), or a hybrid approach (e.g., SMOTE-ENN) to prevent the model from simply learning to predict the majority class (healthy controls). This omission is a critical methodological flaw that invalidates the reported performance metrics (AUCs).

5.2 Unsatisfactory Model Performance and Selection: While the subtype models show higher AUCs than the binary model, the performance for Subtype 1 (AUC 0.785) and Subtype 2 (AUC 0.817) is modest and may be inflated due to the data imbalance issue. Furthermore, the reliance on a single algorithm (CatBoost) is not well-justified. A robust study would involve testing and optimizing a suite of models, including potentially more powerful deep learning architectures designed for tabular data (e.g., FT-Transformer, GANDALF), to ensure the best possible model is selected.

Minor Concerns:

1. Missing Methodological Details: Figure 2 presents visualizations using UMAP and t-SNE. However, these dimensionality reduction techniques are not described anywhere in the Methods section.

2. Ambiguity of Control Group: It is not explicitly clear whether the same healthy control (HC) group was used for all comparisons against the different depression subtypes in the various analyses (GWAS, machine learning, etc.). This should be clarified.

Reviewer #3: The study employed various methods, including biological assessment, cohort study, and machine learning analysis, to provide valuable insights into specific biomarkers of depression. Providing data on depression subtype and its implications for intervention is valuable and practical. the study used large sample of UK biobank data for analysis which made it more rigorous study.

Some points should be considered for further clarification as follows:

1. In the introduction section, the field of metabolomics study has been compared with binary, categorical, and diagnostic. It would be more appropriate to discuss a dimensional approach to diagnosis and treatment, comparing it with the main approach of the current study.

2. In the introduction ref #12, a study on the UK Biobank sample is mentioned. The biomarkers achieved an AUC diagnostic score of 0.6-0.7, which is insufficient for accurate diagnosis. How it could be justified and explained

3. Why did the "methodology" section come after the discussion section?

4. What exactly is the meaning of "depression": minor, major, dysthymia …. Type of depression? Why has PHQ-2 instead of PHQ-9 been used for case finding? The enrolled cases may not be considered as "major depressive disorder". Major depressive disorder, as mentioned by the article, is a heterogeneous disorder, which should not be considered an exact binary diagnosis by using precise diagnostic criteria.

5. This study helps to make a subtype of depressive disorder and has clinical implications; however, it does not add value to accurate diagnosis, as written in lines 273-274. How could it be justified?

6. "...This suggests that future research could develop streamlined diagnostic models for depression utilizing TRL and different lipoprotein components…" 281-282 LINES, again, do not explain the biomarker for diagnosis of depression. TRL may be involved in many normal and abnormal biological processes, it may just provide a feature of MDD. As mentioned in ref 33, the metabolic network associated with specific depressive symptoms may have intervention implications.

**Do you want your identity to be public for this peer review?** For information about this choice, including consent withdrawal, please see our Privacy Policy

Reviewer #1: No

Reviewer #2: No

Reviewer #3: No

**Figure resubmission:****Reproducibility:** To enhance the reproducibility of your results, we recommend that authors of applicable studies deposit laboratory protocols in protocols.io, where a protocol can be assigned its own identifier (DOI) such that it can be cited independently in the future. Additionally, PLOS ONE offers an option to publish peer-reviewed clinical study protocols. Read more information on sharing protocols at https://plos.org/protocols?utm_medium=editorial-email&utm_source=authorletters&utm_campaign=protocols

---

## [Decision Letter · Decision Letter 1]

20 Nov 2025

Towards Precision Psychiatry: Metabolomics Identifies Three Biological Subtypes of Depression

PDIG-D-25-00377R1

Dear Dr Liu,

We are pleased to inform you that your manuscript 'Towards Precision Psychiatry: Metabolomics Identifies Three Biological Subtypes of Depression' has been provisionally accepted for publication in PLOS Digital Health.

Best regards,

Hadi Ghasemi

Academic Editor

PLOS Digital Health

**Additional Editor Comments (if provided):**

**Reviewer Comments (if any, and for reference):**

Reviewer's Responses to Questions

**Comments to the Author**

Reviewer #3: All comments have been addressed

Reviewer #4: All comments have been addressed

Reviewer #5: All comments have been addressed

publication criteria?

Reviewer #3: Yes

Reviewer #4: Yes

Reviewer #5: Yes

3. Has the statistical analysis been performed appropriately and rigorously?

Reviewer #3: I don't know

Reviewer #4: Yes

Reviewer #5: Yes

4. Have the authors made all data underlying the findings in their manuscript fully available (please refer to the Data Availability Statement at the start of the manuscript PDF file)?

Reviewer #3: No

Reviewer #4: Yes

Reviewer #5: Yes

5. Is the manuscript presented in an intelligible fashion and written in standard English?

Reviewer #3: Yes

Reviewer #4: Yes

Reviewer #5: Yes

Reviewer #3: (No Response)

Reviewer #4: (No Response)

Reviewer #5: Minor comments:

Page 96 (Line 286-288): Please provide reference(s) for the following: Specifically, depressive symptoms may promote behavioral changes that exacerbate metabolic dysfunction, while metabolic alterations may in turn aggravate depressive symptoms through disruptions in energy homeostasis and neuroendocrine regulation.

Page 117 (Line 686): It is mentioned as Figure length, please change it to Figure Legends

Please define all abbreviations at first use (TRL, GlycA, MUFA, etc.)

**Do you want your identity to be public for this peer review?** For information about this choice, including consent withdrawal, please see our Privacy Policy

Reviewer #3: No

Reviewer #4: No

Reviewer #5: None
